# Mitochondrial HMG-Box Containing Proteins: From Biochemical Properties to the Roles in Human Diseases

**DOI:** 10.3390/biom10081193

**Published:** 2020-08-16

**Authors:** Veronika Vozáriková, Nina Kunová, Jacob A. Bauer, Ján Frankovský, Veronika Kotrasová, Katarína Procházková, Vladimíra Džugasová, Eva Kutejová, Vladimír Pevala, Jozef Nosek, Ľubomír Tomáška

**Affiliations:** 1Department of Genetics, Faculty of Natural Sciences, Comenius University in Bratislava, Ilkovičova 6, Mlynská dolina B-1, 842 15 Bratislava, Slovakia; veronika.vozarikova@uniba.sk (V.V.); frankovsky1@uniba.sk (J.F.); katarina.prochazkova@uniba.sk (K.P.); vladimira.dzugasova@uniba.sk (V.D.); 2Institute of Molecular Biology, Slovak Academy of Sciences, Dúbravská cesta 21, 845 51 Bratislava, Slovakia; nina.kunova@savba.sk (N.K.); jacob.bauer@savba.sk (J.A.B.); veronika.kotrasova@savba.sk (V.K.); Eva.Kutejova@savba.sk (E.K.); vladimir.pevala@savba.sk (V.P.); 3Department of Biochemistry, Faculty of Natural Sciences, Comenius University in Bratislava, Ilkovičova 6, Mlynská dolina CH-1, 842 15 Bratislava, Slovakia; jozef.nosek@uniba.sk

**Keywords:** mitochondria, mitochondrial nucleoid, HMG-box protein, Abf2, TFAM, DNA-binding, mitochondrial disease

## Abstract

Mitochondrial DNA (mtDNA) molecules are packaged into compact nucleo-protein structures called mitochondrial nucleoids (mt-nucleoids). Their compaction is mediated in part by high-mobility group (HMG)-box containing proteins (mtHMG proteins), whose additional roles include the protection of mtDNA against damage, the regulation of gene expression and the segregation of mtDNA into daughter organelles. The molecular mechanisms underlying these functions have been identified through extensive biochemical, genetic, and structural studies, particularly on yeast (Abf2) and mammalian mitochondrial transcription factor A (TFAM) mtHMG proteins. The aim of this paper is to provide a comprehensive overview of the biochemical properties of mtHMG proteins, the structural basis of their interaction with DNA, their roles in various mtDNA transactions, and the evolutionary trajectories leading to their rapid diversification. We also describe how defects in the maintenance of mtDNA in cells with dysfunctional mtHMG proteins lead to different pathologies at the cellular and organismal level.

## 1. Introduction

According to the widely accepted endosymbiotic theory [1,2], mitochondria are descendants of a singular merger between an archaeal and an α-proteobacterial cell [3,4,5]. Although the nature of the selection pressure favoring this symbiosis is still being discussed [6,7], it is clear that during the more than 1.0–1.9 billion years of the relationship [8], most of the genes of the original endosymbiont were either lost or transferred into the nucleus. Yet, with the exception of highly specialized mitochondria such as the mitosomes and most hydrogenosomes, a handful of protein-coding genes as well as part of the genes for the RNAs involved in their translation, have been retained in the organelle. The reasons why mitochondria keep their own genome are the subject of ongoing debate [9,10]. Although this question is still unanswered, there is a vast amount of literature on how the mitochondrial genome is maintained, expressed, and segregated during organellar and cellular division (for review see [11,12,13]). This review focuses on one particular issue related to both mtDNA maintenance and inheritance, how the DNA is compacted into higher-order structures.

## 2. Mitochondrial DNA Forms Higher-Order Structures Called Mitochondrial Nucleoids

One characteristic feature of mitochondrial genetics is the polyploid nature of the mitochondrial genomes. Although the individual molecules are several orders of magnitude shorter than the nuclear chromosomes, the total amount of mtDNA per cell can reach substantial lengths. For example, standard haploid yeast *Saccharomyces cerevisiae* cells contain on average about 50 copies of a 75–86 kbp long mtDNA molecule [14,15,16,17]. A mature human oocyte contains more than 150,000 copies of a 16.6 kbp mtDNA [18,19]; if the mtDNA circles were stacked one atop the other, they would form a column 300 μm high and almost 3 m of mtDNA would need to be packed into a cell only 100–200 μm across [20]. Even human cells with relatively lower mtDNA copy numbers, such as fibroblasts (2000 mtDNA copies per cell [21]) and leucocytes (ca. 360 mtDNA molecules per cell [22]), face a challenge in fitting their mitochondrial genomes into a limited organellar volume. 

To circumvent this problem, the mtDNA is compacted into condensed nucleo-protein structures called mitochondrial nucleoids (mt-nucleoids) [23,24,25,26,27,28,29,30,31,32,33,34,35,36,37,38]. Considering the diameter of the mt-nucleoids in *S. cerevisiae* (0.3–0.6 μm) [24], yeast mtDNA must undergo a compaction of roughly three orders of magnitude. The size, shape, number of mt-nucleoids per cell (50–100) as well as the number of mtDNA copies per nucleoid (3.9–20) in *S. cerevisiae* depend on its physiological conditions [23]. Mammalian mt-nucleoids contain a relatively small number of mtDNA molecules (e.g., mouse fibroblasts contain about 3 mtDNA molecules per nucleoid) and are more diffuse compared to their yeast counterparts [25,39,40]. 

The mt-nucleoid protein composition has been examined in detail especially for yeast and mammalian cells [25,27,34,41,42,43,44,45,46,47,48,49,50]. It has been shown that mitochondria harbor morphologically distinct subpopulations of nucleoids associated with the inner membrane [40,51], which may differ in their involvement in various types of mtDNA transactions [52]. Somewhat unexpectedly, in addition to the proteins involved in mtDNA replication, transcription, recombination (e.g., DNA and RNA polymerases, topoisomerases, DNA helicases), and translation (e.g., ribosomal protein Mnp1, RNA helicases), mt-nucleoids were also shown to contain a substantial number of proteins involved in metabolism, membrane transport, cytoskeletal dynamics, and protein quality control [25,48,53]. Although the roles of individual components in the maintenance of mt-nucleoids are unknown in most cases, the great diversity of proteins associated with the nucleoid shows that there is an intimate connection between DNA transactions and the other biochemical activities taking place in mitochondria. On the other hand, the major mt-nucleoid components are DNA-binding proteins dedicated to mtDNA compaction. Indeed, all mt-nucleoids examined have one major group of DNA-packaging proteins which contain 1–2 so-called high-mobility group (HMG)-box domains. These proteins, termed mitochondrial HMG-box containing proteins (mtHMG proteins for short) are the primary subject of this review.

## 3. Setting the Stage: The Identification of HMG-Box Proteins in Yeast and Human Mitochondria

The first representative of this large and heterogeneous family of proteins was identified by Caron et al. in yeast by biochemical means [54]. Using the DNA-cellulose chromatography setup employed for the purification of the *Escherichia coli* protein called HU (known to be the principal *E. coli* chromosome packaging protein [55]), the authors identified a lysine-rich, thermostable ~20 kDa polypeptide which they named HM (for histone-like in mitochondria). Using *petite* mutants lacking mtDNA, they inferred that the protein is encoded by a nuclear gene. The abundance of HM was great enough to enable it to organize the structure of the mtDNA. By measuring the introduction of negative superhelical turns in relaxed DNA in the presence of topoisomerase they showed that HM, similar to *E. coli* HU, forms compact structures with DNA. Comparison of the molar amino-acid composition of HM with the HU, HMG2, and histone proteins yielded ambiguous results, so it was impossible to infer a homology to any of these protein classes. Yet, in hindsight it is astounding how this pioneering study correctly described most of the principal biochemical properties of the protein. 

Naturally, there was one essential missing piece, the amino-acid sequence of HM. In the 1980s, the reports on HM were limited to resolving whether or not it represented the yeast H1 histone (it does not) [56,57] and to hints that HM belongs to a group of HMG proteins [56]. This was unequivocally demonstrated by Diffley and Stillman [58] by cloning the gene. Originally aiming for proteins recognizing replication origins (autonomously replicating sequences, ARS) in *S. cerevisiae*, they purified two ARS-binding factors, Abf1 and Abf2 [59]. Whereas Abf1 was demonstrated to be a *bona fide* ARS-binding factor [59,60], Abf2 was shown to be primarily localized to the mitochondria [58]. Later it was found that the genome of *S. cerevisiae* encodes a paralogue of Abf2, Ixr1, that serves as a transcriptional repressor regulating hypoxic genes during normoxia [61]; it also participates in recognizing DNA cross-links [62]. The *ABF2* and *IXR1* genes are products of whole-genome duplication [63], yielding HMG-box proteins localized to the mitochondria and nucleus, respectively. Analogously in mammals, a testis-specific HMG-box protein was shown to be a nuclear isoform of mitochondrial transcription factor A (TFAM; see below) [64], yet it plays a role in the spermatid nucleus during the replacement of histones with nucleoprotamines [65].

Analysis of its amino-acid sequence revealed that, in addition to a 26-residue N-terminal mitochondrial targeting signal [58], Abf2 contains two tandem HMG-boxes, presumably mediating its DNA binding, either of which is sufficient to stabilize mtDNA in vivo [66]. Based on the sequence of these HMG-boxes, Abf2 was classified as an HMG-1 box/HMGB protein [58,67,68] (there are six additional HMG-box containing proteins in *S. cerevisiae*: Hmo1, Hmo2 (Nhp10), Nhp6A, Nhp6B, Rox1, and Ixr1). The amino-acid composition of Abf2 and its ability to supercoil DNA in vitro both resembled the findings of Caron et al. [54]; subsequently, Abf2 and HM were shown to be encoded by the same gene [58,69]. Their differences in sequence indicate that although both yeast Abf2 and *E. coli* HU (and other bacterial nucleoid-associated proteins [70]) participate in the packaging of DNA into nucleoids, they do not share a common ancestor. It also implies that the protein involved in DNA packaging in the original α-proteobacterial endosymbiont was replaced by an HMG protein from the host. This is in contrast to chloroplasts where the packaging of the organellar genome is mediated by an HU-like protein [71]. The involvement of Abf2 in mtDNA maintenance was demonstrated by the analysis of a mutant lacking a functional copy of the *ABF2* gene. When grown on media containing a fermentable carbon source, *abf2*^−^ cells rapidly lost mtDNA and became respiratory deficient. The *abf2*^−^ strains also exhibit other mutant phenotypes (https://www.yeastgenome.org/locus/S000004676#phenotype), such as sensitivity to osmotic stress [72], but these are mostly related to secondary defects in mtDNA maintenance rather than to the direct participation of Abf2 in the corresponding cellular processes. When spores formed by *ABF2*/*abf2*^−^ diploids were transferred directly onto plates containing only a nonfermentable carbon source (glycerol), the *abf2*^−^ haploid cells were able to grow and maintain their mtDNA, indicating that *S. cerevisiae* possesses an Abf2-independent mechanism of mtDNA packaging (see below) [58]. A follow-up study provided insights into the biochemical characteristics of Abf2 [73]. It (i) confirmed the ability of Abf2 to introduce negative supercoils into relaxed double-stranded (ds) DNA, (ii) showed that Abf2 prefers supercoiled rather than relaxed or linear DNA substrates, and (iii) demonstrated that Abf2 binding to DNA is nonrandom and possibly mediated by the phased distribution of short stretches of poly(dA), which prevent Abf2 binding. In total, this suggested that Abf2 might bind preferentially to mtDNA regions that are important for replication, transcription, or recombination. The involvement of Abf2 in mtDNA packaging was also demonstrated by complementation studies where the instability of mtDNA in *abf2*^−^ cells was partially suppressed by the ectopic expression of *E. coli* HU [69]. Furthermore, the ability of the yeast nuclear HMG protein Nhp6A (when fused to the mitochondrial targeting sequence of Abf2) to functionally replace Abf2 indicated that the function(s) of Abf2 in mitochondria may be similar to the function(s) of nuclear HMG proteins [66].

At the same time as Diffley and Stillman were identifying Abf2 as the yeast mtHMG protein, David Clayton’s laboratory was studying a mammalian protein that promoted mtDNA transcription in vitro, originally called mammalian mitochondrial transcription factor 1 (mtTF1, mtTFA) and later renamed mitochondrial transcription factor A (TFAM) [74,75]. The cloning and sequence analysis of the TFAM-encoding gene revealed that, similar to Abf2 in yeast, it is an HMG-box containing protein [76,77]. Its role in transcription initiation has been extensively studied and was found to be quite complex; it is dependent on the corresponding promoter as well as on additional proteins [78,79].

Comparing the DNA-binding properties of human TFAM and *S. cerevisiae* Abf2 revealed several similarities [80]. Biochemical experiments indicated that both proteins bind and bend DNA molecules and actively participate in mtDNA compaction into mt-nucleoid-like structures [29,81,82,83]. Although TFAM and Abf2 share some common characteristics, Abf2 does not seem to act as a transcription factor in *S. cerevisiae* mitochondria [84,85]. It was shown that the addition of the 29-residue C-terminal tail of human TFAM to Abf2 was sufficient to convert the latter into a transcription activator in vitro [86] indicating that it is the absence of the tail that makes Abf2 incompetent to regulate mtDNA transcription; the mitochondrial transcription factor role in yeast mitochondria is performed by the Mtf1 protein [84]. (Of note, the *Drosophila* homologue of mammalian TFAM is also dispensable for transcription [87]). Despite this difference, both human and mouse TFAM were able to complement *abf2*^−^- associated phenotypes in *S. cerevisiae,* indicating that the functions of these proteins overlap [88,89]. Interestingly, it was recently reported that *S. cerevisiae* mitochondria are able to stably maintain mouse mtDNA when they also contain TFAM [90]. The discovery of the functional conservation of mtHMG proteins was very important because it implied that studies on the HMG-box proteins of various eukaryotes can be instrumental not only for understanding their evolution, but also for providing important insights into the general principles of mtDNA maintenance.

## 4. Genetic Studies on mtHMG Proteins Reveal Their Role in Regulating mtDNA Copy Number and Mt-Nucleoid Morphology and Dynamics

The fact that cells lacking Abf2 rapidly lose their mtDNA when grown on glucose [58] was not surprising considering that Abf2 was proposed to be the principal mtDNA-packaging protein. However, there were two issues that were the subject of intensive experimental effort: (i) The relative stability of mutant derivatives of mtDNA in *abf2*^−^ mutants and (ii) the retention of the wild-type mitochondrial genomes in cells lacking the *ABF2* gene when grown on nonfermentable carbon sources.

First, in *S. cerevisiae,* mitochondria can harbor wild-type (*rho*^+^) mtDNA, carrying a full complement of the genes required for cellular respiratory competence, or they can possess its deletion variants (*rho*^−^), containing amplified fragments of mtDNA that are stably propagated. Initial studies clearly showed that Abf2 was essential for maintaining *rho*^+^ mtDNA; however, Zelenaya-Troitskaya et al. [91] found that the *rho*^−^ genomes remained relatively stable in *abf2*^−^ cells. Moreover, they showed that a moderate (2-3-fold) increase in *ABF2* expression yielded a 50–150% increase in mtDNA copy number in both *rho*^+^ and *rho*^−^ cells and that the increase in mtDNA copy number in these cells accelerated nuclear DNA replication, cell proliferation, and mitigated the Sir2-dependent repression of genes, suggesting that mitochondrial processes have an active role in the control of cell division [92]. This crosstalk between the nuclear and mitochondrial genomes is reciprocal. Increasing ribonucleotide reductase activity by overexpressing *RNR1* not only results in increased mtDNA copy number [93], but also partially rescues the mtDNA depletion in the *abf2*^−^ strain [94]. Interestingly, the Abf2 paralogue Ixr1 is involved in regulating *RNR1* expression [95].

While modest *ABF2* overexpression leads to an increase in the amount of mtDNA, when its transcription is driven by a strong *GAL1* promoter, the result is rapid loss (in 2–4 generations) of *rho*^+^ mtDNA but only slow mtDNA loss in *rho*^−^ strains [91]. The different effect on maintenance of *rho*^+^ vs. *rho*^−^ mtDNA in *abf2*^−^ cells could be explained by the participation of Abf2 in the recombination-mediated replication of *S. cerevisiae* mtDNA. Whereas Abf2 would facilitate replication by binding to recombination intermediates in wild-type mitochondria, the *rho*^−^ mtDNA genomes often contain tandem repeats that have an enhanced propensity to recombine independently of Abf2 and thus could stably propagate in its absence. This model was supported by studies demonstrating (i) the preferential binding of Abf2 to various types of recombination intermediates in vitro [96]; (ii) the suggested interaction of Abf2 with the Holliday junction resolvase Cce1 [97] and genetic interaction with the recombinase Mhr1 [98]; (iii) the recombination-dependent mechanism of replication and segregation of yeast mtDNA [99,100,101,102,103]; (iv) the negative effect of an absence of Abf2 on the level of mtDNA recombination in vivo [91,104]; and (v) an increase in recombination intermediates in cells overexpressing *ABF2* [104]. The engagement of Abf2 in recombination may explain in part the defects of mtDNA partitioning within zygotes lacking a functional *ABF2* gene [105], although this phenotype can also be due to the involvement of Abf2 in the segregation of mtDNA, which is independent of recombination. 

The second problem requiring an explanation was the ability of *abf2*^−^ cells to retain their genome when grown on a nonfermentable carbon source. A series of studies from Ron Butow’s laboratory provided some clues to this enigma. An analysis of mt-nucleoids from *abf2*^−^ cells revealed that the pattern of mtDNA staining differs from that of wild-type cells [106]. In permeabilized mitochondria, some regions of mtDNA became hypersensitive to DNase I digestion, while other loci remained unaffected [106]. This indicated that in addition to Abf2, there are other mt-nucleoid components that play a role in compacting mtDNA and that these proteins can participate in maintaining the mt-nucleoids in cells lacking Abf2.

The search for proteins capable of replacing Abf2 in mtDNA maintenance was initiated by screening a yeast episomal library for genomic DNA fragments able to prevent the loss of mtDNA from *abf2*^−^ cells grown on glucose. The screen resulted in the identification of *ILV5* as a multicopy suppressor of the *abf2*^−^ phenotype. *ILV5* encodes an acetohydroxy acid reductoisomerase, which catalyzes a step in branched-chain amino-acid biosynthesis. For efficient suppression, a 2–3-fold increase in the *ILV5* gene copy number was required. Moreover, the stability of mtDNA in *abf2*^−^ cells was increased on media lacking amino-acids, accompanied by an increased Gcn4-dependent expression of *ILV5* [107]. In addition, it was shown that the parsing of mtDNA into nucleoids is regulated by the general amino-acid control pathway and could be separated into one or more activities affecting the recombination of mtDNA and an activity of *ILV5* that controls the organization of mtDNA molecules in nucleoids [108]. It was also observed that the increased number of nucleoids resulting from the activation of the general amino-acid control pathway dramatically increases the transmission of mtDNA, suggesting that this pathway operates on mtDNA organization to increase mtDNA transmission under starvation conditions. On the other hand, the involvement of Ilv5 protein in mtDNA stabilization did not depend on the functioning of the branched-chain amino-acid biosynthesis pathway, indicating bifunctionality of the Ilv5 protein. The amino-acid residues responsible for the catalytic activity and mtDNA maintenance reside in distinct regions of the protein [109]. Although it was shown that Ilv5 binds DNA in vitro [110], how exactly Ilv5 promotes mtDNA compaction remains a mystery.

Mitochondrial nucleoids undergo rapid changes not only when Abf2 is absent, but also under various physiological conditions. Therefore, it was of interest to assess the possible involvement of Abf2 in nucleoid dynamics in vivo. To this end, Kucej et al. [43] performed an *in organello* ChIP-on-chip assay and demonstrated that Abf2 binds to most of the mitochondrial genome with a preference for GC-rich sequences. Under respiring conditions, mt-nucleoids formed open nucleo-protein structures with a lower ratio of Abf2 to mtDNA. The bifunctional nucleoid proteins Hsp60 (a chaperone) and Ilv5 were recruited to mt-nucleoids during glucose repression and amino-acid starvation, respectively. Using co-immunoprecipitation experiments, the authors showed that Aco1 (aconitase), Ald4 (aldehyde dehydrogenase), Idh1, Idh2 (subunits of isocitrate dehydrogenase), and Kgd1 (a subunit of α-ketoglutarate dehydrogenase) are the major protein partners of Abf2 in vivo [43], further supporting the role of bifunctional metabolic enzymes in the maintenance of mt-nucleoids in yeasts [34] and highlighting the general phenomenon of the multitasking or “moonlighting” of proteins originally thought to have dedicated functions in cellular metabolism [111,112]. 

Another screen identifying multicopy suppressor was based on the selection of *abf2*^−^ strains containing an episomal plasmid carrying a fragment of genomic DNA able to support growth on glycerol media at 37 °C [69,113]. Subsequently, three genes were identified. The first, *SHM1* [114], was later shown to encode a mitochondrial GDT/GTP transporter and renamed *GGC1* [115]. The deletion of *GGC1* in an *abf2*^−^ background resulted in synthetic lethality on glycerol, probably due to the combined negative effect of the absence of Abf2 on mtDNA stability and the deficient nucleic acid and protein synthesis caused by an ineffective mitochondrial supply of guanine nucleotides [114]. The second multicopy suppressor, *YHM2*, also encodes an inner mitochondrial membrane carrier protein (for citrate and oxoglutarate [116]) which co-purifies with mt-nucleoids and exhibits DNA-binding properties in vitro [117]. Finally, the overexpression of *TIM17,* which encodes a component of the inner mitochondrial membrane import channel, prevents the complete loss of mtDNA in *abf2*^−^ cells [118]. Conversely, the overexpression of *ABF2* was able to rescue the *petite*-negative phenotype of some *TIM17* mutants [119]. The fact that increased levels of inner membrane proteins such Ggc1, Yhm2, or Tim17 can rescue cells from defects caused by the absence of Abf2 is intriguing. It is known that Abf2 is part of a large (~900 kDa) complex containing the DNA helicase Pif1 and DNA polymerase γ (Mip1) [120] and that there are two distinct subpopulations of mt-nucleoids associated with different regions of the mitochondrial membrane [52] (this is similar to some mammalian cell types, which exhibit distinct subpopulations of mt-nucleoids differing in the amount of TFAM [45,46,121]). Nevertheless, a direct link between Ggc1, Yhm2 and Tim17, and the membrane attachment of the mt-nucleoid is still lacking.

A breakthrough in understanding how the mt-nucleoid is preserved in the absence of Abf2 came from a study demonstrating that aconitase (Aco1) is required for stabilizing mtDNA in *abf2*^−^ cells [47]. When grown on glycerol, *abf2^−^* cells may retain their mtDNA concomitantly with elevated HAP complex-dependent *ACO1* expression under respiratory conditions [122]. The mt-nucleoid functions of Aco1 are independent of aconitase catalytic activity: The introduction of point mutations in the Fe-S cluster required for the aconitase enzymatic activity in the Krebs cycle did not result in the impairment of its ability to stabilize mtDNA in *abf2*^−^ cells [47]. The DNA-binding activity of Aco1 also resides in a distinct region [123]. Such bifunctionality has also been described for the cytosolic form of mammalian aconitase, which oscillates between enzymatic and RNA-binding forms based on the assembly and disassembly of the [4Fe-4S] cluster [124,125]. The involvement of Aco1 in mtDNA packaging in *S. cerevisiae* also indicates that mt-nucleoids exist in different states depending on the metabolic condition of the cells. Whereas Abf2 is essential for mtDNA maintenance under glucose repression, in de-repressed cells with robust oxidative metabolism or in response to retrograde signals, mtDNA packaging may also involve Aco1 [126,127]. It was shown that Aco1 binds to mtDNA (with a preference for GC-rich sequences, similar to Abf2), protects it from an excessive accumulation of point mutations and ssDNA breaks and suppresses the reductive recombination of mtDNA [123]. The participation of aconitase in the formation of the mt-nucleoid not only explains the ability of *abf2*^−^ cells to retain their mtDNA when grown on glycerol [58], but also provides clues about the mechanisms involved in mt-nucleoid remodeling as part of a strategy for adjusting mtDNA maintenance to changes in cellular metabolism [48,128]. 

Another interesting observation on the Abf2-independent mechanism of mtDNA maintenance resulted from an investigation into the cellular responses to the inhibition of the processing peptidase cleaving the mitochondrial N-terminal targeting sequence. It was shown that retaining the N-terminal sequences on the imported proteins makes them insoluble and triggers a specific type of unfolded protein response (mtUPR). Depletion of Abf2 under such conditions was compensated for by the relocalization of the nuclear HMG-box transcription factor Rox1 to the mitochondria, where it binds mtDNA and ensures its maintenance and gene expression [129]. 

Genetic studies were also instrumental for determining the role of mammalian TFAM in mtDNA maintenance in vivo. Whereas heterozygous TFAM^+/^^−^ mice have a 34 ± 7% [130] or ca. 50% [131] reduction in mtDNA copy number, homozygous knock-outs (TFAM^−/−^) die in mid-gestation [130]. Manipulation of the levels of TFAM causes changes in mtDNA copy number and defects in mtDNA segregation [132,133,134]; the removal of TFAM using a conditional knock-out system (*cre-loxP*) resulted in the depletion of mtDNA and mitochondrial transcripts as well as in severe respiratory chain deficiency [130,135,136,137,138,139]. Based on these studies, it was clear that TFAM performs several important functions related to mtDNA packaging, replication, transcription, and segregation and thus is an essential player in the regulation of mtDNA maintenance in mammalian cells [83].

## 5. mtHMG Proteins Are Rapidly Evolving

Comparing the biochemical properties and the in vivo roles of Abf2 and mammalian TFAM reveals that some mtHMG protein functions are shared, but many others seem to be species-specific. In contrast to the extremely conserved canonical core of the histones involved in nuclear DNA packaging [140], the amino-acid sequences of the mtHMG proteins are highly divergent even between species within the same taxonomic group, such as ascomycetous yeasts [49,50,58,141,142,143,144,145], making it difficult to identify them from a simple genomic search. The only common features seem to be the presence of at least one HMG-box for mediating mtDNA binding and a cleavable N-terminal signal peptide for mitochondrial import (Figure 1 and Appendix A). 

For example, the *Kluyveromyces lactis* Abf2 homologue was first identified as a component of purified mt-nucleoids [142] and was later found to exhibit only ~30% sequence identity to *S. cerevisiae* Abf2 [50]. Investigating the mt-nucleoids from several species of *Saccharomyces*, *Pichia,* and *Williopsis* (the latter two now classified as *Cyberlindnera*) did reveal the presence of Abf2-like DNA-binding proteins, but these did not cross-react with antibodies raised against *S. cerevisiae* Abf2, which again highlight the fast evolution of the corresponding ancestral genes [143,146,147].

Considering the affinity of Abf2 for recombination intermediates, it was of interest to search for mtHMG proteins in yeasts with linear mitochondrial genomes, such as *Candida parapsilosis,* whose mitochondrial telomeres are maintained by a recombination-dependent mechanism involving the generation and rolling-circle replication of telomeric repeats [150,151,152]. Purified mt-nucleoids from *C. parapsilosis* contained Gcf1, a protein exhibiting some biochemical similarities to *S. cerevisiae* Abf2, and the corresponding gene complemented the mtDNA stability defect of the *S. cerevisiae abf2^−^* mutant. In contrast to Abf2, an in silico analysis of Gcf1 predicted the presence of a coiled-coil domain and two HMG-boxes, one of which is weakly conserved (Figure 1 and Appendix A), suggesting that it represents a novel type of mtHMG protein [49]. Gcf1 from *C. parapsilosis* has orthologues in *Candida albicans* and a closely related species from the genus *Debaryomyces* [144]. Depletion of *C. albicans* Gcf1 results in a 3000-fold decrease in *GCF1* mRNA levels, which is correlated with a substantial decrease in the number of both mtDNA copies and recombination intermediates. The absence of Gcf1 did not result in the loss of cell viability even though *C. albicans* and *C. parapsilosis* are *petite*-negative species unable to grow without a functional mitochondrial genome. This suggests that, similar to *S. cerevisiae* Abf2, there is a back-up mechanism for mtDNA packaging and maintenance [144].

Another type of mtHMG protein, Mhb1, was found in the yeast *Yarrowia lipolytica*. The protein was identified as the major component of purified mt-nucleoids and was found to compact DNA in vitro [145]. Although *Y. lipolytica* is a strictly aerobic yeast [153], deletion of the Mhb1-encoding gene did not result in loss of viability. Additionally, the mutant exhibits clear differences in mt-nucleoids (Figure 2) accompanied by an increased sensitivity to ethidium bromide, a DNA intercalating agent known to inhibit the synthesis of mtDNA and to make it prone to degradation [154,155]. The mutant had fewer copies of mtDNA and mtDNA-derived transcripts. Interestingly, its respiratory characteristics and growth under most of the conditions tested were indistinguishable from those of the wild-type strain. On the other hand, the level of mtDNA-encoded proteins in the mutant was similar to the wild-type cells indicating that the cells are able to circumvent the potential imbalance between the subunits of the respiratory chain encoded by the nuclear and mitochondrial genomes [145]. 

The existence of compensatory mechanism(s) for dealing with decreased levels of mtDNA or mtDNA-derived transcripts was also indicated by studies of *S. cerevisiae abf2*^−^ mutants and Gcf1-deficient *C. albicans* cells [91,144]. Various explanations of its still enigmatic molecular nature have been suggested [145]. The translational plasticity of human mitochondrial ribosomes has been shown to contribute to preserving a balance between the nuclear and mitochondrial-encoded respiratory complex subunits. Specifically, defects in the assembly of respiratory chain complexes caused by an insufficient supply of subunits translated by the cytosolic ribosomes result in the arrest of mitochondrial translation [156]. Although mechanistically different, such regulatory crosstalk systems have also been described in yeast, showing that the regulation of mitochondrial and cytosolic translation is coordinated [157]. In addition, the accuracy of yeast mitochondrial translation is monitored by the cytosolic proteostasis system in a manner that shares many characteristics with the metazoan mitochondrial unfolded protein response [158]. 

So far, no mtHMG protein has been identified in the fission yeast *Schizosaccharomyces pombe*. Its genome encodes a *S. cerevisiae* Ixr1 homologue, Cmb1, which has been shown to participate in mismatch repair [159] and to colocalize with mitochondria [160]; bioinformatics tools indicate that it should localize to the mitochondria [161] and that it possesses a cleavable targeting sequence (as predicted by mitochondrial protein targeting software MitoProt II; [162]). However, so far no differences between nucleoids in a *cmb1*^−^ mutant compared to the wild-type *S. pombe* cells have been reported. There are other fission yeast HMG proteins with putative mitochondrial localization (such as an Nhp6 homologue), but their involvement in maintaining mt-nucleoids has not been investigated yet. Perhaps the proteomic analysis of purified mt-nucleoids that is underway in the Miyakawa laboratory [23] will pinpoint the protein responsible for mtDNA packaging in fission yeasts. 

In the filamentous fungus *Podospora anserina*, a mtHMG protein was identified as the product of a gene that suppressed premature cell death caused by the accumulation of deletions in mtDNA [141]. Although the protein was not characterized biochemically, it was shown to reside in the mitochondria. In addition, it was the first example of a mtHMG protein combining the AT-hook (a DNA-binding motif consisting of a conserved proline-arginine-glycine-arginine-proline core sequence) [163] and HMG-box DNA-binding domains. In *Aspergillus nidulans*, the protein HmgB is located primarily in mitochondria, but also exhibits nuclear localization. In addition to a canonical HMG-box at the C-terminus, it also contains two structurally related domains called Shadow-HMG-boxes. An HmgB deletion strain exhibited a decrease in conidial and ascospore viability, probably due to the importance of HmgB in maintaining mtDNA in spores [164]. Furthermore, it was also shown that HmgB plays a role in cellular protection against oxidative stress agents, although it is possible that this role is associated with its nuclear functions [165].

A divergent mtHMG protein termed Glom was identified in the condensed mt-nucleoids of the true slime mold, *Physarum polycephalum* [166]. In addition to two HMG-boxes at its C-terminus, Glom also has a lysine-rich region with a proline-rich domain in its N-terminal half, and all three domains seem to participate in DNA-binding. Interestingly, the lysine-rich region alone was sufficient for its intense mtDNA condensation in vitro as well as for condensing the *E. coli* chromosome into nucleoid structures in vivo. On the other hand, the proline-rich region was necessary for keeping the nucleoid accessible to the transcription machinery and the HMG-boxes were required for complementing the defects associated with the absence of the HU protein. Thus, Glom is a distinct type of mtHMG protein which employs three DNA-binding domains and, in coordination with the *S. cerevisiae* Mgm101 homologue Glom2, plays a complex role in the organization and dynamics of the *P. polycephalum* mt-nucleoid [166,167].

The kinetoplast DNA (kDNA) of unicellular flagellates from the order Kinetoplastida is composed of a complex, mixed population of maxi- and minicircles [168], and presents a particularly challenging mitochondrial genome for packaging. In *Crithidia fasciculata*, several basic polypeptides, called kinetoplastid-associated proteins (KAPs), were identified by biochemical means [169,170], and some of them were shown to play a role in kDNA maintenance [171]. In *Trypanosoma brucei*, the KAP6 protein, containing two HMG-boxes, was shown to be involved in kDNA replication [172] and to possess DNA-binding properties similar to other mtHMG proteins [173].

Although mt-nucleoids have been studied in higher plants [174,175], their principal mtDNA packaging protein(s) remain elusive. The fact that no plant mtHMG protein has been identified to date may be either due to its distant relationship to known mtHMG proteins or because mtDNA packaging in plants does not require a mtHMG protein [176,177]. The resolution of this issue will require the detailed proteomic analysis of purified plant mt-nucleoids.

The rapid evolutionary sequence diversification of mtHMG proteins may be due to the fact that amino-acid substitutions outside the HMG-boxes are generally neutral. A pairwise comparison of the amino-acid sequences of the other protein components of the mitochondrial nucleoids of *C. albicans* and *C. parapsilosis* showed that they are more similar than those of their corresponding HMG proteins [49], yet mtHMG proteins retain their ability to form functional complexes with the latter, indicating that their faster diversification do not affect functional protein–protein interactions [178]. Alternatively (or concomitantly) changes in mtHMG proteins may reflect their species-specific biochemical roles in mtDNA maintenance and segregation, or possibly differences in mtDNA base composition and topology. To explore this possibility we compared the biochemical properties of the mtHMG proteins from three distantly related yeast species: *S. cerevisiae* (Abf2), *Y. lipolytica* (Mhb1), and *C. parapsilosis* (Gcf1) and observed several differences [96], but two major areas of commonality. We found that (i) all three proteins exhibit relatively weak binding to intact dsDNA, as demonstrated by the fact that Abf2 and Mhb1 bound to this type of substrate only at very high protein:DNA ratios while Gcf1 showed only negligible binding; and that (ii) all three proteins exhibited a high preference for recombination/replication intermediates such as Holliday junctions and replication forks. These observations highlight the involvement of yeast mtHMG proteins in the maintenance and compaction of mtDNA in vivo. The relatively high affinity of Gcf1 for these types of DNA structures suggests that the protein may be actively involved in the maintenance of mitochondrial telomeres that rely on the recombination-dependent formation of extragenomic circular DNA (telomeric circles or t-circles) [151,152,179]. Mammalian TFAM was also shown to have a high affinity for recombination intermediates [180], thereby contributing to the peculiar organization of the mtDNA in complex junctional networks in human heart muscle as well as in mouse and human brain tissue [181,182].

## 6. mtHMG Proteins Protect mtDNA against Damage and the Accumulation of Mutations 

mtDNA molecules are primary targets of reactive oxygen species (ROS) and therefore must be protected against oxidative damage. mtHMG proteins appear to represent one of the first lines of defense against ROS. 

Spontaneous *petite* colony formation in the *abf2^−^* strain was partially suppressed by malonic acid treatment, which decreases ROS formation by the electron transport chain (ETC). ROS generation by the ETC therefore likely contributes to mitochondrial disfunction in *abf2^−^* cells. The *abf2^−^, ntg1^−^* double null mutant (which also lacks the DNA *N*-glycosylase and apurinic/apyrimidinic lyase Ntg1), also exhibits a synthetic *petite* phenotype if untreated. Importantly, the negative effect of *ABF2* deletion is not due to compromised recombination, which indicates that *S. cerevisiae* possesses recombination-independent mechanisms to cope with oxidative mtDNA damage [183]. Deletion of either *ABF2* or *NTG1* results in the activation of Rad53 checkpoint signaling and the reduction of the mitochondrial ROS-mediated adaptive extension of chronological lifespan [184].

Another study assessed the types of mutations accumulating in *abf2*^−^ strains and found an increase in the frequency of frameshifts and direct-repeat mediated deletions with no change in the rate of mtDNA point mutations. Treatment of *abf2^−^* cells with UV light resulted in a relative increase in respiratory-deficient mutants when compared to the wild-type [185]. These results highlight the different architecture of the mt-nucleoids in *abf2*^−^ cells, which renders their mtDNA more vulnerable to genomic instability. A propensity to higher rates of mtDNA mutations was also observed for the *Y. lipolytica mhb1*^−^ strain [145]. This indicates that, although the absence of mtHMG proteins is well tolerated in the short term even in strictly aerobic species, the increased frequency of mutagenic events accumulating within the mitochondrial genome results in lower fitness overall, and such strains are thus subject to negative selection. TFAM was also involved in DNA repair where it was shown to preferentially bind to mtDNA damage hot spots in rat cells, thus making them less accessible to DNA repair factors. The import of p53 under oxidative stress conditions was suggested to facilitate the weakening of the affinity of TFAM for the damaged sites thereby enabling their repair [186,187,188]. Yoshida et al. [189] observed a significant increase in cisplatin-damaged DNA-binding by TFAM upon interaction with p53 in vitro, however.

## 7. On the Mechanism of mtDNA Compaction by mtHMG Proteins 

Abf2 binds DNA with modest cooperativity and affinity. The reported K_D_ varies between 0.04–1.5 μM [190,191,192], although it was suggested that the affinity of Abf2 for DNA changes by an order of magnitude during its compaction (from 0.28 to 2 μM), indicating that DNA accessibility to Abf2 decreases significantly with increasing compaction [193]. Several studies have determined the number of Abf2 molecules per cell with a median abundance of 15,000 ± 7000 [194] and a range from 3307 [195] to 51,132 [196]. These large differences are partly due to different growth conditions, to the use of different strains, and to different means of quantification. This needs to be considered when extrapolating the results of in vitro studies to the situation in vivo. With the total mitochondrial volume in a haploid cell being about 0.2–1.5 μm^3^ [15,197], the concentration of Abf2 would range from 27.5–425 μM (for 0.2 μm^3^) to 4–55 μM (for 1.5 μm^3^), making the lowest estimated concentration higher than the greatest apparent K_D_ estimate. 

Using optical and atomic force microscopy (AFM) it was shown that Abf2 is able to compact dsDNA linear molecules. Based on a fast Abf2 off-rate (k_off_ = 0.014 ± 0.001 s^−1^) the packaging of DNA was relatively weak and the forces stabilizing the condensed DNA–protein complex were small (<0.6 pN). The visualization of individual complexes by AFM revealed 190-nm structures that were loosely packaged compared to nuclear chromatin, indicating that they are accessible for transcription and replication and, at the same time, more vulnerable to damage [191]. However, such an arrangement does not pose a threat to yeast mitochondrial genome integrity, as other mitochondrial proteins participate with Abf2 in mtDNA stabilization in vivo [34]. A follow-up study showed that Abf2 binding induces sharp bends (of about 78°) in the DNA backbone for both linear and circular DNA and that at a high Abf2 concentration, the DNA suddenly collapses into a tight nucleo-protein complex. Based on these results, a “bent-worm-like chain model” was created in which the introduction of bends into the DNA is sufficient to promote compaction without the need for super-twisting [192]. It may be noted here that structural studies on bacterial HU, which is not an HMG-box protein, demonstrated that it also bends the DNA by 106–124° [198,199]. This suggests that bending is a fairly common means for compacting DNA. However, both studies used a distorted DNA template containing three T–T mismatches and four unpaired Ts. A more recent structure using native DNA shows that HU may actually straighten rather than bend its DNA substrates [200].

The essential information for a detailed understanding of the mechanism of mtDNA compaction was provided by the determination of the 3D structures of mtHMG proteins. HMG proteins normally contain one or two HMG-box domains. Structurally, an HMG-box domain typically contains three α-helices arranged in an L-shape. The short leg of the L is comprised of two short, anti-parallel α-helices (α1 and α2) while the longer leg consists of α3 together with an elongated N-terminal segment of around 6–7 residues. Structures of HMG-box domains in complex with DNA show that the protein binds to the DNA minor groove through the concave surface of the L with widening and flattening of the minor groove and pronounced bending of the DNA substrate. This bending is stabilized by several polar and non-polar interactions together with the intercalation of non-polar residues from either or both α1 and α2, which disrupt the DNA base-pair stacking. In the Abf2–DNA complex [201] residues Phe-51 (from α1 of HMG-box 1) and Ile-124 (α1 of HMG-box 2) intercalate into the DNA. Important polar contacts were identified for residues Lys-44 and Arg-45 (box 1) and Lys-117 and Lys-118 (box 2). DNA backbone-interacting residues included Tyr-50, Tyr-53, Arg-77, Trp-81, and Lys-89 from box 1 and Phe-123, Tyr-126, Ile-150, Trp-154, and Lys-162 from box 2. Overall, each Abf2 molecule bound to two different DNA substrates using its two HMG-box domains, thereby “stapling” them together (Figure 3). This study also confirmed that the protein displays a distinct “phased-binding” at DNA sequences containing poly-adenine tracts (A-tracts) [58,73]. The two crystal structures of Abf2 in complex with mtDNA-derived fragments bearing A-tracts showed that each Abf2 HMG-box induces a 90° bend in the contacted DNA, causing an overall U-turn. Furthermore, it was demonstrated that an N-terminal flag and α-helix are crucial for mtDNA maintenance. They promote the initial binding of the protein to DNA and facilitate the subsequent interactions with the HMG-boxes which are accompanied by DNA bending. The structure also indicated that Abf2 might be excluded from the A-tracts because these tracts have a narrow minor groove which might make them inaccessible for Abf2-binding [201]. Since the mtDNA of *S. cerevisiae* is rich in A-tracts, this property could be important for setting the overall nucleoid architecture. On the other hand, there are presently no structures of either Abf2 bound to GC-rich sequences or to recombination intermediates, both of which are its preferred substrates [43,96]. Such studies could shed more light on one additional paradox: It has been shown that for efficient compaction in vitro, the ratio of Abf2 to mtDNA should be one protein molecule per ~10–20 bp [58,201]. In addition, 50–100 molecules of ~80 kbp mtDNA per yeast cell [23] yields 2–8 × 10^5^ binding sites, yet even the largest estimate is about 5 × 10^4^ molecules of Abf2 per cell [196], i.e., almost one order of magnitude lower than that needed to reach the optimal ratio. It is possible that the compaction of mtDNA is achieved by the combined effect of Abf2 and other mt-nucleoid proteins, but it is also likely that GC-rich regions or recombination intermediates play a more important role in the compaction [96].

TFAM is present in mammalian mitochondria in a ratio of about 1000 molecules per mtDNA (or 1 TFAM per 15–18 bp of mtDNA) making it abundant enough to coat the entire mitochondrial genome [39]. The affinity of TFAM for nonspecific DNA substrates, as determined by surface plasmon resonance (4 nM) [29] and other methods [83], seems to be higher than that of Abf2 and it is even higher for promoter sequences (0.16–1.6 nM), indicating a distinct difference in their function and possibly reflecting the role of TFAM as a transcription factor. TFAM could also contribute to the regulation of mtDNA transactions through G-quadruplex binding. The in vitro affinity of TFAM for certain G-quadruplex structures, which might form in the GC-rich regions of human mitochondrial genomes, was higher than for the corresponding B-DNA [202].

Similar to Abf2, TFAM contains two HMG-boxes, but now they are separated by a longer, 30-residue linker, followed by a C-terminal extension involved in regulating transcription. Moreover, similar to Abf2, TFAM binds to the DNA minor groove using its HMG-box domains, but in this case the two domains bind to the same DNA substrate. Both domains bind to the convex surface of the resulting bent DNA by passing the 30-residue α-helical linker through the bend (Figure 3). This results in a conformation in which the N-terminal, short legs of the HMG-box domains are oriented towards one another across the complex. The linker helix is positively charged, which helps to offset the electrostatic strain introduced by bringing the negatively charged phosphate backbone atoms of the DNA closer to one another. TFAM HMG-box 1 intercalates Leu-58 and partially Tyr-57 (both from α1) while Thr-77, Thr-78, and Ile-81 from α2 make important contacts. HMG-box 2 intercalates Leu-182 (which corresponds to Ile-81 from box 1) while Asn-163 (topologically equivalent to Leu-58) hydrogen-bonds to two T nucleotides, imparting a shear rather than intercalating and Tyr-162 partially intercalates. Trp-88 from box 1 and Trp-189 from box 2, together with a number of other residues from the α3 helices, stabilize the phosphate backbone through polar contacts and positive charges [203,204]. More detailed experiments showed a stepwise binding mechanism where DNA-binding to HMG-box 1 initially bends the DNA into a V-shape reducing the distance between HMG-boxes and eventually facilitating the binding of HMG-box 2. Each HMG-box induces a 90° bending of the minor groove and inserts two leucines (Leu-58 from HMG-box 1 and Leu-182 from HMG-box 2) into two sites separated by 10 bp, thereby completing the typical U-shaped conformation [204]. TFAM functions as a homodimer and generally binds DNA in a sequence-non-specific manner with the exception of sequence-specificity to sites upstream (−15 to −35) of the heavy strand (HSP1) and light strand (LSP) promoters [205]. When bound to the LSP, the U-turn bending of TFAM allows the C-terminal tail, which recruits the transcription machinery, to approach the initiation site [206,207]. The crystal structures of TFAM bound to HSP1 showed that it binds in the opposite direction compared to LSP thereby explaining the different modes of transcription activation at these two sites. While it is unnecessary for DNA bending and transcriptional activation, TFAM dimerization was suggested to be important for mtDNA compaction, possibly by promoting DNA looping [208]. In particular, all TFAM–DNA complexes solved to date feature an interface between the α3 helices of HMG-box 1 from two different complexes. The convex outside surfaces come into contact, with each α3 paired in an anti-parallel orientation. The interface buries a surface area of approximately 1180 Å^2^ and is characterized by a series of polar and electrostatic interactions involving especially residues Lys-95, Tyr-99, Glu-106, Glu-112, and Arg-116. Mutants which disrupt this interface are still able to bind DNA and initiate transcription, but cannot compact DNA as efficiently as the wild-type protein [208]. A more recent study using a combination of biochemical methods and super-resolution and electron microscopies showed that a mutant TFAM unable to dimerize still compacted DNA. It also revealed that TFAM-mediated nucleoid formation in vitro is a multistep process. It is initiated by TFAM binding single DNA duplexes as beads on a string followed by the TFAM molecules bridging two DNA duplexes to form loops [209]. A cooperative manner of DNA binding was proposed, meaning that the first TFAM binding event influences the second—a characteristic that increases the affinity of TFAM for DNA and drives DNA compaction to completion [29]. Removal of the C-terminal tail only slightly decreased non-sequence-specific DNA binding, and the X-ray crystal structure of HMG-box 2 revealed unusual features for an HMG-box, including interactions of the HMG-box with other regions of TFAM [210].

## 8. Regulation of mtHMG Proteins 

Despite sharing no sequence homology with eukaryotic nuclear histones or HMG-box proteins, bacterial nucleoid-associated proteins (NAPs) are primarily responsible for the dynamic spatial organization of the bacterial nucleoid, the homeostasis of DNA supercoiling, and global gene regulation. In general, NAPs are highly conserved within specific bacterial families, some even highly conserved among all prokaryotes, and all bacterial species possess at least one NAP [211,212]. Bacterial NAPs provide higher-order superstructures that link gene expression and the architecture of the bacterial genome. Their DNA-binding properties allow distant domains to be brought into proximity or two distinct parts of the DNA to be bridged, which significantly influences life-dependent DNA transactions [212]. It is also known that the structure of the nucleoid and global transcription patterns are both influenced by environmental conditions [213]. A more recent *E. coli* study confirmed that nucleoids undergo visible changes during cell growth with the DNA more tightly compacted during the stationary phase than exponential growth; this has been partly attributed to changes in the expression levels of genes encoding the NAPs that were more abundant during the growth phase [214].

Similar to their bacterial counterparts, mt-nucleoids are sensitive to changes in the levels of active mtHMG proteins. As discussed above, a decreased amount of mtDNA-bound Abf2 was found to increase the transcriptional activity of mt-nucleoids in *S. cerevisiae* [43] whereas the strong overexpression of Abf2 leads to a rapid loss of mtDNA [91]. These results underline the importance of regulating the levels of mtHMG proteins for maintaining cellular functions. 

High-throughput transcriptomic analyses showed several conditions that change in response to changes in *ABF2* expression (https://www.yeastgenome.org/). For example, it was shown that *ABF2* transcription is under the control of Hcm1, whose overexpression or deletion leads to altered Abf2 levels accompanied by changes in mitochondrial biogenesis [215]. Similarly, expression of the *ABF2* gene is decreased in cells lacking its paralogue Ixr1 when they are exposed to hypoxia [61]. A whole chromatin immunoprecipitation analysis revealed that the *ABF2* promoter interacts with the transcription factors Fhl1 and Rsf1 when cells are exposed to heat [216]. Despite this, the relevance of the regulation of *ABF2* expression on a transcriptional level has not been investigated in sufficient detail. The ratio of Abf2 to mtDNA changes more as a result of fluctuations in mtDNA copy number rather than changes in *ABF2* expression [43]. The possibility that Abf2 might be regulated post-transcriptionally emerged from a study demonstrating that about half the *ABF2* mRNA transcripts are associated with mitochondria, and thus could potentially provide a direct source of Abf2 to the organelle [217]. Furthermore, Abf2 levels increased in cells treated with rapamycin [218], a selective inhibitor of mTOR (the mechanistic target of rapamycin), a protein kinase that acts as a central integrator of the nutrient signaling pathways [218]. In contrast, deletion of the *TOR1* gene resulted in increased mitochondrial respiration through the stimulation of mtDNA translation without affecting the mtDNA copy number [219]. 

The intracellular levels of TFAM must be finely tuned to prevent the dysregulation of mtDNA transactions that can result in various pathologies (see Section 9; Table 1 and Appendix A). Expression of the gene encoding TFAM is transcriptionally controlled by peroxisome proliferator-activated receptor-γ coactivator-1α/nuclear respiratory factor-1 [220,221]. In patients with glioma, KLF16 serves as a key regulator of glioma cell proliferation by binding to the TFAM promoter, leading to the reduction of TFAM expression [222]. TFAM can also be downregulated as a result of hypermethylation of the promoter, which can be induced by cigarette smoke, as in the case of chronic obstructive pulmonary disease [223].

In mouse spermatocytes and elongating spermatids, specific transcripts encode a protein isoform that is imported into the nucleus [224]. Testis-specific isoforms were also found in human germ cells, but they were shown to be transported into the mitochondria [225]. The authors showed that TFAM and mtDNA are present in high levels in undifferentiated male germ cells but in low levels or are absent in differentiated spermatocytes. Testis-specific transcripts exhibited the opposite pattern, suggesting that they may play a role in downregulating mitochondrial biogenesis during spermatogenesis.

In addition, the TFAM mRNA is subject to complex post-transcriptional regulation (Figure 4), including stabilization by RNA-binding proteins [226] and various microRNAs [227,228,229,230]. For example, microRNA-494 (mir494) was shown to downregulate TFAM expression in proliferating myoblasts by binding to 3′-UTR within the mRNA sequence. On the other hand, exercise in mice resulted in decreased expression of mir494 and elevated expression of TFAM accompanied by the stimulation of mitochondrial biogenesis [231], although mir494 was also shown to act upstream of TFAM via a transcriptional coactivator in some cases [232,233]. The TFAM pre-mRNA also undergoes alternative splicing resulting in the production of two major TFAM isoforms: The full-length polypeptide and a version lacking the last 32 amino-acids, which corresponds to most of the second HMG-box [234,235,236]. Both isoforms seem to form the active mitochondrial transcriptional complex [237]. However, the interplay between these two (and other minor putative, tissue-specific) isoforms in vivo has not been studied in detail.

In addition to regulation at the levels of gene expression, translation and functional substitution, post-translational modification (PTM) also plays an important role by providing an extra layer of flexibility for protein regulation according to actual need (Figure 4). In eukaryotes, the most studied and the most influential of these proteins are histones, which can be acetylated, methylated or ubiquitinylated and thus influence nuclear DNA replication, chromatin stability, the degree of DNA compaction and the level of DNA transcription [238,239]. Not so widely studied but still fairly common are PTMs in prokaryotes [240,241,242,243]. Mass spectrometry data from 11 independent proteomic studies revealed over 100 unique PTMs in the four most abundant *E. coli* nucleoid-associated proteins (HU, H-NS, IHF, and FIS) emphasizing the potential impact of these modifications on their in vivo functions [244]. It is worth noting, however, that a large proportion of the PTMs identified in this study were acetylated lysines. Only a few protein-modifying enzymes have been discovered in bacteria, therefore it seems likely that most of this acetylation occurs non-enzymatically [244].

To date, several PTMs have been described for mitochondrial HMG-box proteins. For *S. cerevisiae* Abf2, it was found that the N-terminal extended segment of the first HMG-box is phosphorylated by a cAMP-dependent protein kinase (PKA) in vitro resulting in a decrease in its DNA-binding activity [190]. An Abf2 mutant lacking the corresponding phosphorylation sites exhibited a severe defect in the regulation of mtDNA content during glucose repression in vivo. It was shown that the first HMG-box of TFAM is also a substrate for PKA in human mitochondria and that the modified protein has a decreased binding to DNA as well as reduced transcriptional activity [245]. Another TFAM phosphorylation site, found in the second HMG-box, is modified by the extracellular signal-regulated protein kinase ERK1/2. A phosphorylation-mimicking mutant showed a reduced affinity for the LSP, although binding to non-specific mtDNA sequences was not affected [246]. 

PTMs could regulate mtHMG proteins at the levels of protein–protein interactions, substrate binding, and stability. Moreover, it has been shown that phosphorylated TFAM is a substrate for the ATP-dependent protease Lon [245]. *S. cerevisiae* Abf2 was also demonstrated to be subject to Lon-mediated degradation. When bound to DNA, TFAM and Abf2 are protected from Lon-mediated proteolytic degradation thus providing a possible mechanism for adjusting the stoichiometry of mtHMG proteins to the available mtDNA substrate [247]. TFAM was shown to be phosphorylated in cells treated with rapamycin, although its effect on TFAM DNA-binding activity was not tested [248]. Therefore, the potential involvement of Abf2 and TFAM in TOR signaling is currently unclear.

In addition to phosphorylation, other post-translational modifications can alter the DNA-binding properties of mtHMG-box proteins. For example, mammalian TFAM was shown to be acetylated in vivo [249] and later in vitro studies demonstrated that acetylated TFAM has a significantly lower dsDNA-binding affinity [250]. In contrast, a more recent study showed that acetylation significantly decreased the DNA-unwinding ability of TFAM, while its DNA-binding ability was largely unaffected [251]. It is possible that different combinations of acetylated lysines have different effects on the biochemical properties of TFAM. In clear cell renal cell carcinoma, the ability of TFAM to interact with mtDNA is impaired when it is acetylated at Lys-154 and therefore the protein deacetylase SIRT3 regulates the TFAM function and mitochondrial biogenesis [252]. Additionally, acetylated Abf2 [253] and succinylated forms of both Abf2 and TFAM have been detected in vivo (in HeLa cells for TFAM) [254], although the biological significance of these modifications has not been assessed.

Recently, it was demonstrated that phosphorylation and acetylation have different effects on the kinetics of TFAM binding to DNA [250]. In general, phosphorylation introduces a negative charge and adds steric bulk, which likely causes an electrostatic repulsion with the phosphate backbone of DNA. On the other hand, acetylation neutralizes the positive charges of lysine side-chains and adds a similar steric bulk. The authors proposed that lysine acetylation reduces DNA-binding due to a lower on-rate, whereas the serine phosphorylation results in a decreased on-rate and an increased off-rate [250]. Another interesting finding is that the PTMs of eukaryotic mitochondrial proteins may also occur outside of the mitochondria prior to their import into the organelle. For example, TFAM may undergo ubiquitination [255,256,257], which, in the retina cells of diabetic patients, prevents its entry into the mitochondria and causes retinal cell dysfunction and subsequent sight loss [258]. Furthermore, TFAM can be modulated by O-linked β-N-acetylglucosamine glycosylation [259], although its effect on the activities of TFAM remains unclear. It is likely that TFAM and other mtHMG proteins undergo additional modifications to provide means for the epigenetic regulation of mtDNA maintenance [38] (Figure 4).

## 9. Involvement of TFAM in Cellular Pathologies

Considering the roles of TFAM, it is not surprising that defects in its functions result in pathologies collectively named mitochondrial DNA maintenance defects (MDMDs). These represent a group of diseases caused by pathogenic variants in several nuclear genes involved in mtDNA maintenance [260]. These pathologies can be caused by impaired mtDNA synthesis leading to qualitative (mtDNA mutations) and quantitative (mtDNA depletion [261]) defects in mtDNA (Table 1 and Appendix A).

The MDMDs are often accompanied by changes in the cellular levels of TFAM. Pioneering studies in mice demonstrated the importance of TFAM in mtDNA homeostasis (see Section 4) [130]. Furthermore, heterozygous mice lacking one copy of the TFAM gene were more prone to metastasis in an intestinal cancer model [262], and heterozygous cells produced more inflammatory cytokines, most likely due to higher levels of mitochondrial stress signaling [131]. Increased levels of cytokines were shown to participate in chronic inflammation accompanying accelerated aging in T cells with dysfunctional mitochondria caused by TFAM deficiency [263]. Tissue-specific knockout in the skeletal muscle and heart produced some fraction of viable offspring, however, these individuals developed a dilated cardiomyopathy similar to the human Kearns-Sayre syndrome during their life [135,138,264]. Mouse muscles depleted of TFAM exhibited decreased levels of Ca^2+^ in the sarcoplasmic reticulum and increased levels of Ca^2+^ in the mitochondria. The latter was explained by an acute stimulation of mitochondrial metabolism, which results in the long-term cell damage observed in these mice [265].

Overexpression of TFAM increases the mtDNA copy number [132] and also causes significant modifications in mtDNA transcription and replication in mt-nucleoids reconstituted in vitro [299] and in cultured human cells [133]. Neurodegenerative model studies showed that the overall mtDNA levels were reduced by ~30% with an even more significant reduction (~50%) of TFAM protein levels. However, in both cell and animal neurodegenerative models (Alzheimer’s, Parkinson’s, and Huntington’s diseases), increasing TFAM levels by mild overexpression or enzyme replacement considerably improved neural function and content [300]. In addition to neuroprotection, a TFAM overexpression-induced increase in mtDNA levels also facilitates a cardioprotection associated with limited mitochondrial oxidative stress or decreased cardiac aging and cardiac remodeling [301,302,303].

The variability in mtDNA copy number among different cancers was reviewed recently by Yuan et al. [304] and is summarized in Table 1. Microsatellite (in)stability was linked to differential mtDNA copy numbers in colorectal cancer [305]. Another example of high variability is in esophageal squamous cell carcinoma. Lin et al. [297] showed an increased copy number in patient samples, while Masuike et al. [298] showed a decreased mtDNA copy number in correlation with malignancy. In some cases (e.g., liver in extrahepatic cholestasis and hepatocellular carcinoma, melanoma cell lines, or prostatic cancer cell lines treated with arsenic), compromised levels of TFAM may result in mtDNA mutations or deletions [267,288,289,295].

TFAM expression does not have the same effect on the prognosis of all types of TFAM-associated cancers. Expression of TFAM at higher levels was associated with longer overall survival time in cases of diffusely infiltrating astrocytomas [286]. Conversely, in cases of epidermoid cancer, normal-small cell lung carcinoma, ovarian cancer, endometrial cancer, and breast cancer patients with TFAM-positive tumors had poorer overall survival [277,279,290,291,293]. Prognosis of disease development was also negatively affected with lower mtDNA copy number in esophageal squamous cell carcinoma and with low expression of KLF16 in glioma cells [222,298]. 

In addition to pathological states caused by altered levels of wild-type TFAM, there are several examples of dysfunctional mitochondria associated with single-nucleotide substitutions within both the coding and noncoding regions of the TFAM-encoding gene (Table 1 and Appendix A). For example, rare variants −91 C→A in 5′-UTR exon 1 and the Ala105Thr missense mutation of TFAM were shown to be involved in the pathogenesis of cardiac hypertrophy [306] and variants rs10826175 (A→G) SNP upstream and rs1937 (G→C) SNP in exon 1 of TFAM were associated with diffuse-type gastric cancer [274]. Genetic variants of TFAM (SNPs rs1049432; rs1937) can have an active role in host response to *Mycobacterium leprae,* causing leprosy, and affect the risk of infection [135,307]. Pro-178 within HMG-box 2 is an important residue for intercalating into the DNA minor groove and creating contacts between TFAM and mtDNA [208]. Manifestations of mtDNA depletion syndrome 15 are caused by a Pro178Leu substitution. The phenotypic manifestations identified include decreased mtDNA copy number and basal respiration, decreased number of nucleoids, and the presence of abnormal nucleoid aggregates. Patients exhibited the neonatal onset of rapidly progressive liver failure, resulting in death in infancy. The primary patient fibroblasts showed increased TFAM mRNA but decreased protein levels, indicating a compensatory mechanism [266]. Possibly, mutant TFAM accumulates inside mitochondria and is recognized and degraded by the Lon protease [245]. 

In a human prostate cancer cell line and in tissue samples of ovarian cancer, TFAM was detected not only in the mitochondria, but also in nuclear chromatin, where it could regulate the expression of nuclear genes in addition to those in the mitochondria [290,308]. This example underlines the pleiotropic effect of TFAM on cellular functions making interpretations of the molecular mechanisms of TFAM-dependent pathologies quite difficult. On the other hand, it makes this small mitochondrial protein and its counterparts in other eukaryotes even more fascinating subjects for future investigation.

In addition to serving as a prognostic marker, TFAM might also be employed as a candidate therapeutic agent. Mitochondrial dysfunction has been reported in various forms of heart failure and TFAM overexpression showed an increased mtDNA copy number and cardioprotective effects against ROS-mediated mitochondrial oxidative stress in transgenic mice [301,309]. A positive effect on survival was observed after myocardial infarction leading to pathological hypertrophy in the case of transgenic mice with TFAM overexpression and even in the cardiac myocytes of a transgenic mouse model with overexpressed recombinant human TFAM [310,311]. TFAM also exhibited therapeutic potential in the reduction of protease expression involved in heart failure [302]. 

The overexpression of TFAM also has a potentially synergistic effect with exercise training in enhancing mitochondrial function in order to prevent skeletal muscle atrophy [312]. TFAM overexpression also has a positive impact on maintaining the mtDNA copy number before decline in the dorsal root ganglia of experimental rat diabetic neuropathy [313]. This is probably due to its role in the SIRT1-PGC-1α-TFAM signaling pathway, which is impaired in diabetic peripheral neuropathy [314].

A TFAM-mediated “ProtoFection” was employed to deliver mtDNA into the mitochondria of cells representing a model of Parkinson’s disease and resulted in the restoration of mtDNA levels accompanied by the activation of mitochondrial biogenesis [315]. Similarly, recombinant TFAM reduced the level of reactive oxygen species and amyloid β accumulation in a cell model of Alzheimer’s disease [316].

Since some cellular pathologies are caused by an increased level of mitochondrial biogenesis, some interventions are aimed at decreasing the level of TFAM. For example, antisense oligonucleotides were used to decrease the expression of TFAM in injured rat carotid arteries [271] and knockdown of TFAM in gastric cancer cells resulted in decreased cell proliferation [274].

## 10. Conclusions 

The symbiosis that gave rise to modern eukaryotic organisms required fine-tuning the coordination between the genomes of the original host and the endosymbiont. This led to an apparatus for maintaining the mtDNA able to protect, replicate, and express its genetic information in response to intra- and extracellular stimuli. The principal players in the mitochondrial genome maintenance are mtHMG proteins that mediate the compaction of mtDNA molecules into mt-nucleoids. The properties and regulators of mtHMG proteins have been adapted for the unique characteristics of the mitochondrial genomes and are pivotal in these transactions. This review provides a comprehensive analysis of mtHMG proteins highlighting the following key points: (1) The mechanism of DNA compaction by mtHMG proteins is relatively conserved and is primarily mediated by the bending of DNA by tandemly arranged HMG-boxes; (2) with the exception of the HMG-boxes, the amino-acid sequences of the mtHMG proteins exhibit faster evolutionary diversification than the other mt-nucleoid protein components; (3) the activity of the mtHMG proteins (and thus indirectly the accessibility of mtDNA to biochemical transactions) is regulated at all levels of gene expression, including transcription activation/repression, alternative splicing, regulation by miRNAs, posttranslational modifications, and proteolytic cleavage; (4) in many (even strictly aerobic) microorganisms, the deletion of a gene for a mtHMG protein is in part compensated for by employing other molecular means of mtDNA compaction; and (5) compromising the functions of the mammalian mtHMG protein TFAM results in a variety of pathological states whose nature depends on the particular mutation and the affected tissue. 

Although research on mtHMG proteins during the last four decades has provided substantial insight into their functions and evolution, there are still important questions that need to be addressed. For example, how does the metabolic state of a cell or modifications to the mtDNA affect the DNA-mtHMG protein interactions? Which additional posttranslational modifications affect mtHMG proteins? What are the effects of these modifications on the structure and stability of mtHMG proteins? Can combining different posttranslational modifications provide the means for producing a heterogeneous intramitochondrial population of mtHMG proteins with distinct DNA- or protein-binding properties? What are the mechanisms of co-evolution of the conserved protein components of the mt-nucleoid and the rapidly evolving mtHMG proteins? How is mtDNA compacted in cells lacking mtHMG proteins? Are there other possibilities for using TFAM as a prognostic tool or a therapeutic target for pharmacological interventions in human diseases with aberrant mtDNA maintenance? Answering these and other related questions will be instrumental for understanding the molecular bases of pathologies linked to the dysfunction of mtHMG proteins and to mitochondria in general.

## Figures and Tables

**Figure 1 biomolecules-10-01193-f001:**
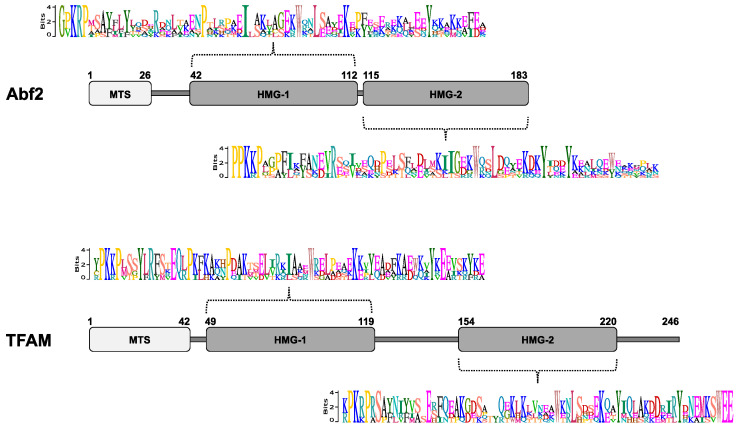
The domain organization of yeast (Abf2) and human (mitochondrial transcription factor A (TFAM)) HMG-box containing proteins (mtHMG) proteins. The amino-acid sequences of the two HMG-box domains of Abf2 (from *S. cerevisiae*, *S. uvarum*, *Candida glabrata*, *Naumovozyma castellii*, *N. dairenensis*, and *Vanderwaltozyma polyspora*) and TFAM (from human, mouse, rat, *Xenopus laevis*, and *Drosophila melanogaster*) were predicted using Simple Modular Architecture Research Tool (SMART) [148] and aligned using multiple sequence alignment software MAFFT (multiple alignment using fast Fourier transform) v7.450 [149]. The sequence logos were created using Geneious v11.1.5 (Biomatters). The mitochondrial targeting sequences (MTS) are based on experimental evidence [58,76].

**Figure 2 biomolecules-10-01193-f002:**
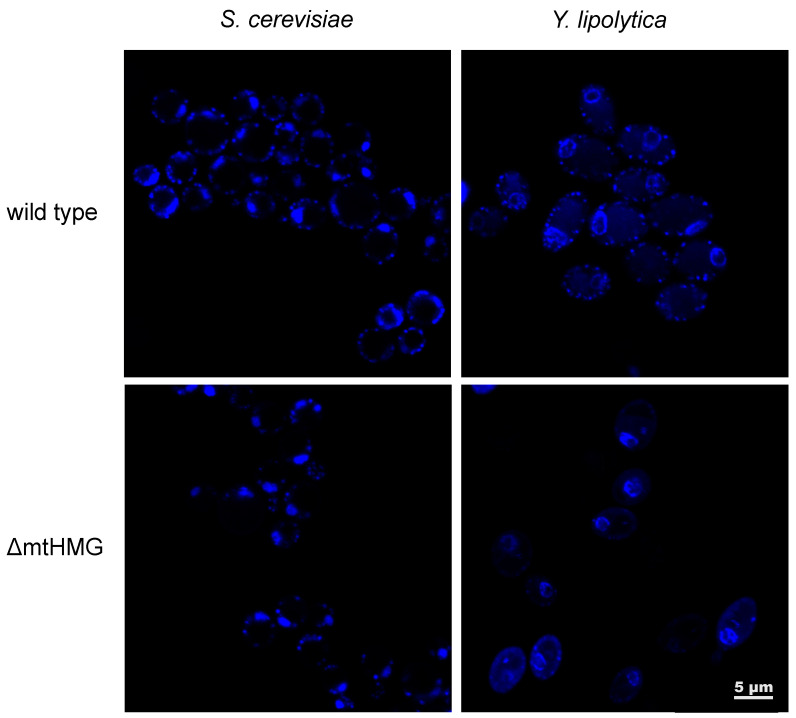
Visualization of DNA in wild-type cells vs. mutants lacking mtHMG protein (ΔmtHMG). Nuclear DNA (large blue dots) and mt-nucleoids (smaller blue spots) were visualized in yeast cells using confocal microscopy (Olympus IX81) after DAPI staining. As in the *S. cerevisiae abf2*^−^ mutant, the deletion of mtHMG genes in *Y. lipolytica,* a strictly aerobic yeast, resulted in a decreased number of mt-nucleoids.

**Figure 3 biomolecules-10-01193-f003:**
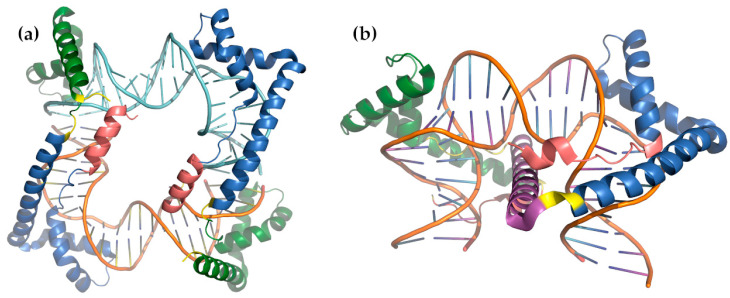
Structures of Abf2 and TFAM in complex with DNA. (**a**) Abf2 in complex with DNA (PDB ID 5JH0; [201]); HMG-box 1 is colored blue, HMG-box 2 is colored green. The two separate DNA molecules are colored differently to differentiate them. The pink α-helices are the additional N-terminal α-helices known so far only from Abf2. (**b**) TFAM in complex with DNA (PDB ID 3TMM; [207]); HMG-box 1 is colored blue, HMG-box 2 is colored green, the 30-residue linking helix is colored magenta. The N-terminal extension is pink.

**Figure 4 biomolecules-10-01193-f004:**
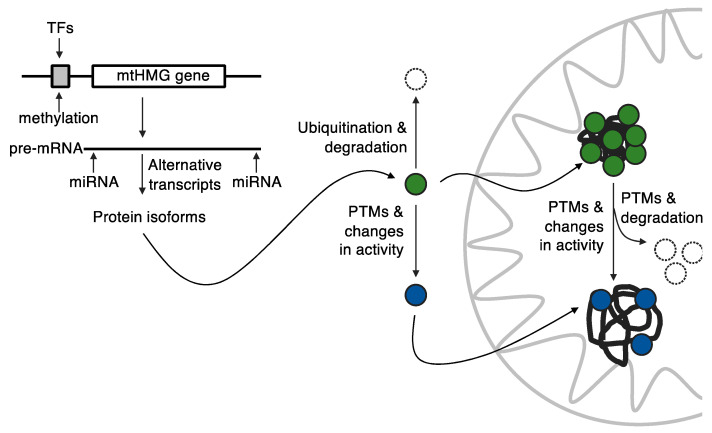
Possible means of regulating mtHMG proteins. The levels of mtHMG proteins can be controlled at the level of transcription by employing specific transcription factors (TFs) as well as by epigenetic markers such as DNA methylation. In the case of TFAM, alternative splicing generates isoforms differing in the number of functional HMG-boxes. Prior to or following their import into mitochondria, mtHMG proteins can undergo various post-translational modifications (PTMs) that can affect their susceptibility to proteolytic degradation, propensity to interact with protein partners (not shown), and DNA-binding activity. As a result, mitochondrial DNA (mtDNA) can be either tightly compacted or more relaxed and thus more accessible to the components of the replication, transcription, recombination, and translation machineries. See text for more details.

**Table 1 biomolecules-10-01193-t001:** Examples of human diseases whose pathologies involve TFAM.

Disease	Model	Variant	mRNA	Protein	mtDNA	Ref.
Mitochondrial DNA depletion syndrome 15	patient	+/+ Pro178Leu	↑ in primary fibroblasts	↓ in primary fibroblasts	↓ copies in liver and skeletal muscle	[266]
Extrahepatic cholestasis	cell line rat, patient	gene disruption	↓ in liver	(↑)↓ in liver	↓ copies mtDNA deletions in liver	[267,268,269]
Cardiomyopathy	mouse	KO in germline	↓	↓	↓copy in +/−depletion in +/+	[130]
Cardiomyopathy	mouse	KO in heart	–	↓ in heart	↓ transcripts in heart	[135,138,264]
Myopathy	mouse	KO in skeletal muscle	–	↓ in muscle	↓ copies ↓ transcripts depletion in muscle	[138,139]
Infantile mitochondrial myopathy	cell line patient	–	↑ ^1^ ↓ ^3^	↓	depletion ^1^,^3^ skeletal muscle	[270]
Carotid artery injury	rat	KD	↑after injury ↓ KD	↑ after injury ↓ KD	↑ mass after injury depletion KD	[271]
Chronic obstructive pulmonary disease (with squamous cell lung cancer)	cell line patient	hyper-methylation of promoter	↓ in lungs ^3^	↓ in lungs, skeletal muscle ^3^	–	[223,272,273]
Gastric cancer	cell line	KD	↓	↓	↓ copies	[274]
Colorectal cancer with microsatellite instability	cell line patient	Leu149Stop frameshift	–	↓	↓ copies	[275]
Colon cancer	mouse	KO +/−	–	↓ in intestinal tissue ^2^	depletion ↓ copy ↓mass	[262]
Colon cancer	cell line patient	↑ miRNA-590-3p	↑	↑	–	[229]
Colon cancer	cell line patient	↓ miRNA-214	↑	↑	–	[276]
Epidermoid cancer/ Colon adenocarcinoma	cell line patient	chemo-therapeutics treatment	↓↑	↓↑	–	[189,277]
Bladder cancer	cell line	↓ miRNA-590-3p	–	↑	–	[230]
Clear cell renal cell carcinoma	cell line patient	KD ^1^	↓ ^1^	↓ ^1^	↓ copies	[278]
Clear cell renal cell carcinoma	cell line	OE SIRT3	↑	↑ ↓K154 acetylation	↑biogenesis	[252]
Non-small cell lung cancer	cell line patient	KD ^1^	↓ ^1^ ↑ ^3^	↓ ^1^ ↑ ^3^	↓ copies ^1^ ↑ copies ^3^	[279]
Lung adenocarcinoma/ Breast cancer	cell line	–	↑in lactic acidosis in long term estradiol treatment	–	↑ copies ↑ mass in lactic acidosis in long term estradiol treatment	[280,281]
Lung adenocarcinoma	cell line	KD, α-irradiation (α-IR)	–	↑ in α-IR	↑ copies in α-IR ≈ copies KD in α-IR	[282]
Lung adenocarcinoma	cell line patient	–	–	↓ ^1^	↓ biogenesis ↓ volume	[283]
Glioma	cell line patient	↓ miRNA-23b	↑ correlation with malignancy	↑ correlation with malignancy	–	[227,284]
Glioma	cell line patient	interaction with KLF16	↓ in KLF16 OE	↓ in KLF16 OE	–	[222]
Glioma	cell line	melatonin treatment	↓	↓	↓ transcripts ≈ copies	[285]
Diffusely infiltrating astrocytoma (a type of glioma)	patient	–	↑ correlation with malignancy	–	↓ copies correlation with malignancy	[286]
Arsenic-induced Bowen’s disease	cell line patient	arsenic exposure ^1,3^ RNA interference ^1^	↑	↑	↑ copies	[287]
Melanoma	cell line patient	–	↓↑	–	mutations	[288]
Prostate cancer	cell line	–	↑ in arsenic exposure	↑ in arsenic exposure	mtDNA point mutation in arsenic exposure	[289]
Cervical cancer	cell line	miRNA-214 OE	↓	↓	–	[228]
Ovarian cancer	patient	–	–	↑ in nuclei orrelation with grade	–	[290]
Endometrial cancer (the estrogen-related type I)	patient	–	–	↑ correlation with stage	↑ copies	[291,292]
Breast cancer (estrogen-positive type)	cell line patient	KD ^1^	↑ ^3^	↑ ^3^	–	[293,294]
Hepatocellular carcinoma	patient	–	–	↑	↓ copies mtDNA deletions	[295]
Esophageal squamous cell carcinoma	cell line patient	KD ^1^	–	↓ ^1^	↑ copies ^3^ ↓ copies ^1^	[296,297]
Esophageal squamous cell carcinoma	cell line patient	KD ^1^	↓ ^1^	↓ ^1^	↓ copy correlation with malignancy ↓ copies ^1^	[298]

↑: Increase; ↓: Decrease; +/+: Homozygous; +/−: Heterozygous; ^1^: Cell line; ^2^: Mouse; ^3^: Patient; KO: Knockout; KD: Knockdown; OE: Overexpression; see Appendix A for more details.

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
