# Peer review of "Mitochondrial HMG-Box Containing Proteins: From Biochemical Properties to the Roles in Human Diseases"

_biomolecules, 2020, doi:10.3390/biom10081193_

Round 1

Reviewer 1 Report

Mitochondrial HMG box proteins (mtHMG proeins) have important roles in compaction of mtDNA, protection of mtDNA against damage, the regulation of gene expression and the segragation of mtDNA into daughter organelles. In this review, the authors first described the history of the discovery of yeast Abf2p and human TFAM. Subsequently, they provided a comprehensive overview of the biochemical properties of the mtHMG proteins, the structural basis of their interaction with DNA, their roles in various mtDNA transactions, the evolutionary trajectories leading to their rapid diversification and the involvement of TFAM in cellular pathologies.

This review adequately covers more than 300 published articles from the past to the recent years. I am convinced that this review will be very helpful to researchers in many areas related to HMG protein. I strongly recommend this manuscript should be published after the minor revision.

minor revision

1, References No. 66 and No.74 are overlapped.

2, In page 9 line 409, does ot → does not

3, In page 9 line 413, nucleoid → nucleoid proteins

4, In page 10 line 429, show → shown

5, In page 10 line 436, stain→ strain

6, In page 11 line 465, ad → and

7, In page 25 line 1052, High Mobility Group → high mobility group

8, In page 26 line 1093, Saccharomyces cerevisiae (italic)

9, In page 26 line 1095, changesin → changes in

10, In page 29 line 1200, Intramitochondrial → intramitochondrial

11, In page 30 line 1247, Pichia jadinii (italic)

12, In page 31 line 1287, Schizosaccharomyces pombe (italic)

13, In page 36 line 1499, Mitochondrial Transcription Factor →mitochondrial transcription factor

14, In page 40 line 1679, Mitochondrial Transcription Factor →mitochondrial transcription factor

Author Response

We thank the reviewer for the positive evaluation of our manuscript and for pointing out several typographical errors. Our corrections are listed below the original reviewer's comments (in bold)

1, References No. 66 and No.74 are overlapped.

The reference has been corrected.         

2, In page 9 line 409, does ot → does not

The typo was corrected.

3, In page 9 line 413, nucleoid → nucleoid proteins

The phrase was changed to "Other protein components of the mitochondrial nucleoid...“

4, In page 10 line 429, show → shown

The typo was corrected

5, In page 10 line 436, stain→ strain

The typo was corrected

6, In page 11 line 465, ad → and

The typo was corrected

7, In page 25 line 1052, High Mobility Group → high mobility group

The reference was corrected

8, In page 26 line 1093, Saccharomyces cerevisiae (italic)

The reference was corrected

9, In page 26 line 1095, changesin → changes in

The reference was corrected

10, In page 29 line 1200, Intramitochondrial → intramitochondrial

The reference was corrected

11, In page 30 line 1247, Pichia jadinii (italic)

The reference was corrected

12, In page 31 line 1287, Schizosaccharomyces pombe (italic)

The reference was corrected

13, In page 36 line 1499, Mitochondrial Transcription Factor →mitochondrial transcription factor

The reference was corrected

14, In page 40 line 1679, Mitochondrial Transcription Factor →mitochondrial transcription factor

The reference was corrected

Reviewer 2 Report

The manuscript presents an extensive review of the mitochondrial HMG proteins family. Their identification, description, functions, and regulation are described as long as something has been published about it.

Considering the amount of work done on these proteins this is a remarkable job and a lot of work. The authors focused mainly on yeasts and mammalian proteins as there are the most studied ,and present the data clearly. I congratulate them for trying to make prefectly understandable something so complex and for some parts still largely unknown and I enjoyed reading this review.

Some minor points which could be clarified are listed below:

line 306, the authors cite ref 49 for "the amino-acid sequences of the mtHMG proteins exhibit an exceptionally high rat of evolutionary divergence" while this is a comparisont between two proteins only. They should be something more general. Same is true line 413 while it is correct line 332

line 349/ » ethidium bromide, an agent known to 
 destabilize mtDNA 
”… This is not very precise concerning the role of ethidium bromide and there are no references.

Lines 359-365. This part is ambiguous and difficult to read for an outsider. Could it be clarified somewhat?

The conclusion is fine but very general and I would have appreciated a larger view. First list the important key points which have been clarified (the review is dense so for no specialists it would be nice to get a simple “take home” message (even if of course this is oversimplified). Then in a second part it would be nice to get some hints about what is the most interesting avenues to proceed in the future from people who really know their subject. Then, their very general few lines are fine.

Author Response

We thank the reviewer for the positive evaluation of our manuscript and below we provide point-by-point responses to his/her suggestions (the original reviewer's comments are in bold):

line 306, the authors cite ref 49 for "the amino-acid sequences of the mtHMG proteins exhibit an exceptionally high rate of evolutionary divergence" while this is a comparisont between two proteins only. They should be something more general.

The sentence was modified to “...the amino-acid sequences of the mtHMG proteins are highly divergent even between species of the same taxonomic group, such as ascomycetous yeasts”, including additional references.

Same is true line 413 while it is correct line 332

The sentence was modified to “Pairwise comparison of the amino-acid sequences of the other protein components of the mitochondrial nucleoids of C. albicans and C. parapsilosis showed that they are more similar than those of their corresponding HMG proteins [49], yet they retain their ability to form a functional complexes with the latter, indicating that the faster diversification of the HMG proteins do not affect functional protein–protein interactions”.

line 349/ » ethidium bromide, an agent known to  destabilize mtDNA ”… This is not very precise concerning the role of ethidium bromide and there are no references.

The sentence was modified to “... ethidium bromide, a DNA intercalating agent known to inhibit the synthesis of mtDNA and to make it prone to degradation” and two original references were included.

Lines 359-365. This part is ambiguous and difficult to read for an outsider. Could it be clarified somewhat?

This part was modified to clarify the text.

The conclusion is fine but very general and I would have appreciated a larger view. First list the important key points which have been clarified (the review is dense so for no specialists it would be nice to get a simple “take home” message (even if of course this is oversimplified). Then in a second part it would be nice to get some hints about what is the most interesting avenues to proceed in the future from people who really know their subject. Then, their very general few lines are fine.

We understand this reviewer’s point and have extensively modified the conclusion according to his/her suggestions.